# WellDunn: On the *Robustness* and *Explainability* of Language Models and Large Language Models in Identifying Wellness Dimensions

**Seyedali Mohammadi[1\*], Edward Raff[1,2], Jinendra Malekar[3],**
**Vedant Palit[4], Francis Ferraro[1], Manas Gaur[1]**
[1]UMBC, MD, USA [2]Booz Allen Hamilton [3]USC, SC, USA [4]IIT, Kharagpur, India
[\*]mohammadi@umbc.edu

## Abstract

Language Models (LMs) are being proposed for mental health applications where the heightened risk of adverse outcomes means predictive performance may not be a sufficient litmus test of a model's utility in clinical practice. A model that can be trusted for practice should have a correspondence between explanation and clinical determination, yet no prior research has examined the *attention fidelity* of these models and their effect on *ground truth explanations*. We introduce an evaluation design that focuses on the robustness and explainability of LMs in identifying Wellness Dimensions (WDs). We focus on two existing mental health and well-being datasets: (a) Multi-label Classification-based MULTIWD, and (b) WELLXPLAIN for evaluating attention mechanism veracity against expert-labeled explanations. The labels are based on Halbert Dunn's theory of wellness, which gives grounding to our evaluation. We reveal four surprising results about LMs/LLMs: (1) Despite their human-like capabilities, GPT-3.5/4 lag behind RoBERTa, and MEDALPACA, a fine-tuned LLM on WELLXPLAIN fails to deliver any remarkable improvements in performance or explanations. (2) Re-examining LMs' predictions based on a confidence-oriented loss function reveals a significant performance drop. (3) Across all LMs/LLMs, the alignment between attention and explanations remains low, with LLMs scoring a dismal 0.0. (4) Most mental health-specific LMs/LLMs overlook domain-specific knowledge and undervalue explanations, causing these discrepancies. This study highlights the need for further research into their consistency and explanations in mental health and well-being.

## 1 Introduction

According to the National Institute of Mental Health ([NIH, 2023](#)), over 20% of US adults have experienced mental illnesses, prompting the government to allocate $280 billion to address unmet

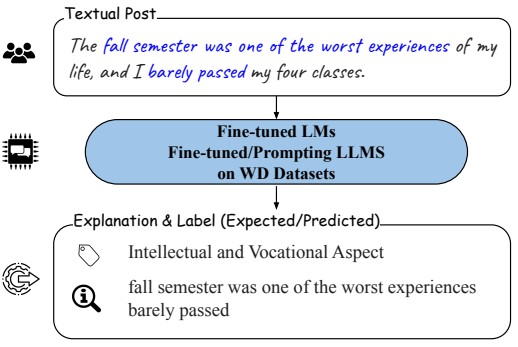

Figure 1: **Motivating Example from WELLXPLAIN dataset**. Expert annotators categorize user posts into four WD classes and justify their choice by highlighting pertinent parts of the text. In LM or LLM classification tasks, the goal is to identify one of the labels (1: Physical, 2: Intellectual and Vocational, 3: Social, 4: Spiritual and Emotional) based solely on relevant cues in the post. The cues are the explanations.

mental health service needs ([White-House, 2023](#)). This highlighted the need to leverage AI (particularly LMs/LLMs) for mental health, as they can potentially decrease costs and increase the accessibility of mental health services. However, vigilance is crucial regarding the potential risk of LMs/LLMs arising from low-confidence predictions and correct predictions with wrong explanations.

Motivated by this longer-term goal of safe deployment of NLP-based mental health systems, we propose evaluation schemes examining the consistency in LM's attention (and LLM's attention where the attention is accessible) with ground-truth explanations[1] and confidence in predictions. Our insight is that a *model's attention in disagreement with physician assessment is unlikely to be accepted, regardless of predictive accuracy*. Indeed, such a scenario implies the model has learned some shortcut or correlative signal instead.

We present an evaluation framework, acronymized as WellDunn, which exam-

---

[1]In this context, we are using 'explanation' that refer to 'text-span explanations' which are tokens/spans of text that are relevant for determining class labels.

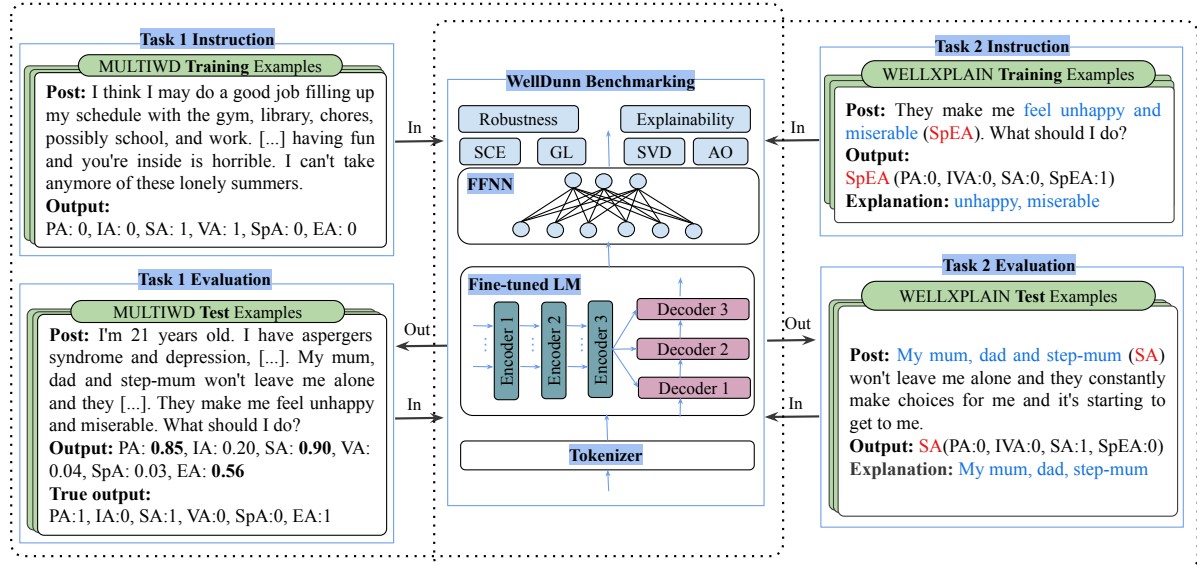

Figure 2: **WellDunn workflow**: MULTIWD task (L) and WELLXPLAIN task (R). The architecture includes shared steps: (1) Fine-tuning of general purpose and domain-specific LMs for extracting data representations, followed by (2) feeding them into a feed-forward neural network classifier (FFNN). Two loss functions assess LMs' robustness: Sigmoid Cross-Entropy(SCE) and Gambler's Loss(GL). Singular Value Decomposition (SVD) and Attention-Overlap (AO) Score assess the explainability. In: Input, and Out: Output. WellDunn Benchmarking Box: This middle rectangle highlights the components of the benchmark system, which includes steps of (1) Fine-tuning and (2) FFNN classifier, as well as Robustness and Explainability components. The Left and right dotted rectangles grouped the components for the MULTIWD and WELLXPLAIN tasks, respectively. In the case of Task 1, the input (text post) is fed into the MULTIWD task, and the model produces an output (prediction) in terms of various WDs like PA, IA, SA, etc. For Task 2, the input (text post) is also fed into the WELLXPLAIN task, which produces output (prediction) along with corresponding explanations. Note that in the instruction (training), we provide both input and output, but in the evaluation (test), we provide the input.

ines **11** LMs/LLMs using two domain-grounded datasets annotated with the causes of deteriorating wellness in individuals. These datasets are important because they consist of user-generated content from individuals expressing signs of depression, bipolar disorder, anxiety, suicide, schizophrenia, or comorbidities caused by the decline in their wellness. The MULTIWD[2] dataset (Sathvik and Garg, 2023) assigns six interconnected Wellness Dimensions (WDs)—Physical, Intellectual, Vocational, Social, Spiritual, and Emotional—to each textual post (crawled from Reddit's posts) based on Halbert L. Dunn's classification (Dunn, 1959; Sathvik and Garg, 2023). This dataset frames the task as a multi-label classification, evaluating LMs/LLMs predictive performance in contexts where WDs are interdependent (Halleröd and Seldén, 2013). The WELLXPLAIN[3] dataset (Garg, 2024; Liyanage et al., 2023) assigns a single WD to each textual post, with annotations explaining the reasons behind the label. Figure 1 presents an example from WELLXPLAIN, where an LM/LLM predicts a WD and offers an explanation,

highlighting the text that captures the model's attention.

**Welldunn Evaluation Criteria:** We utilize traditional evaluation metrics along with supplementary ones, including *SVD rank*, *Attention-Overlap score*, and *Attention Maps*. The *SVD rank* assesses the focus of attention in LMs[4] while the *Attention-Overlap score* measures the extent to which the model's attention aligns with ground truth explanations in the WELLXPLAIN dataset. Figure 2 illustrates the procedure of WellDunn.

**Findings:** Our empirical research into LMs and LLMs for mental health and well-being revealed several key findings: *(a) domain-specific LMs/LLMs performed within 1% of general-purpose models*; on average, general-purpose LMs showed a 1.3% improvement in performance over domain-specific LMs/LLMs. *(b) general-purpose LMs exhibited higher confidence in their predictions compared to domain-specific models*. After retraining four general-purpose and three domain-specific LMs with a confidence-oriented loss function—gambler's loss (a variant of sigmoid cross-

---

[2]https://github.com/drmuskangarg/MultiWD
[3]https://github.com/drmuskangarg/WellnessDimensions/

[4]LLMs' internal machinery is not as transparent as LMs'.

entropy)— general-purpose LMs exhibited 6.3% higher confidence and significantly better attention compared to domain-specific LMs. The decrease in scores is attributed to LMs abstaining from making low-confidence predictions. *(c) general-purpose LMs demonstrated more focused attention than domain-specific LMs, including* LLAMA *and* MEDALPACA. In an inter-model comparison on WELLXPLAIN, LLMs underperformed by 32.5% in MCC compared to vanilla RoBERTa, which also demonstrated higher confidence.

*Takeaway:* These findings challenge assumptions about the efficacy of larger models and the value of fine-tuning in mental health applications. These gaps lead to incorrect and misleading explanations when these models are queried for causes of mental health issues, undermining their reliability and clinical utility. The attention overlap score of 0.0 for LLAMA and MEDALPACA, along with the significant gap between SVD rank and the average length of explanations, supports our inferences and demonstrates significant failures can still occur.

Note: Attention as a medium of explanation is debatable, as inferred from prior works by Bibal et al. (2022); Jain and Wallace (2019) and Wiegreffe and Pinter (2019). However, in these studies, the datasets did not have explicit expert-provided explanations, which can be used to cross-check the overlap between high-attention words and natural language explanations. As in this research, we have a dataset with natural language explanations; we consider attention a medium of explanation.

## 2 Related Work

*AI in Mental Health:* Previous studies in the convergence of AI and mental health concentrated on creating or improving machine learning and deep learning algorithms to identify mental health conditions (MHCs) or assess their severity (Lin et al., 2020; Cao et al., 2020; Lin et al., 2017; Haque et al., 2021). However, minimal attention has been dedicated to ensuring these AI-driven models' robustness and explanatory capabilities. As a result, researchers and practitioners lack insight into whether these models emphasize the correct clinically relevant terms to make decisions and whether they are made with confidence.

To overcome this challenge, efforts have been made to create knowledge-grounded, expert-curated datasets incorporating clinical expertise. These datasets utilize clinical knowledge in vari-

ous forms, such as human experts acting as crowd workers, e.g. Shen et al. (2017), CLPsych by Coppersmith et al. (2015), mental health lexicons (Gaur et al., 2019), and clinical practice guidelines (Gupta et al., 2022; Zirikly and Dredze, 2022). A recent study by Garg (2023) has enumerated 17 classification datasets focused on mental health outcomes, including suicide risk, depression, mental health, stress, and emotion. Various domain-specific and general-purpose LMs have been trained on these datasets. However, the robustness and attention of these models have not been thoroughly examined. This study addresses this gap by adapting WELLXPLAIN's clinically validated explanations for comparative analysis alongside attention mechanisms so that we can test model attention's alignment with a causal determinant.

*Wellness Dimensions:* The severity of MHCs and their cormorbities varies among individuals (Coppersmith et al., 2021). Despite knowledge-grounded datasets, LMs face challenges in generalizing effectively (Harrigian et al., 2020). This difficulty arises from overlooking signs of mental disturbances that can trigger sub-clinical depression and progress to clinical depression over extended periods if left undetected. These signs go beyond the traditional psycholinguistic assessment of natural language, which involves using lexicons like LabMT (Reagan, 2018), ANEW (Bradley and Lang, 1999), and LIWC (Pennebaker et al., 2001). There's a rising interest in using WDs to advance mental health research with LMs (Liyanage et al., 2023). This study is the first to use LMs in mental health, focusing on the model's attention and confidence in predicting WDs.

## 3 Datasets

| Dataset | #Sample | Avg. words/post |
|---|---|---|
| MULTIWD | 3281 | 632 |
| WELLXPLAIN | 3092 | 112 |

Table 1: **Basic statistics of MULTIWD and WELLXPLAIN datasets**: #Sample and Avg. words/post represent the number of samples (each sample includes a post and its six labels) and the average number of words per post respectively.

We utilize two expert-annotated and domain-grounded datasets, MULTIWD (Sathvik and Garg, 2023) and WELLXPLAIN (Garg, 2024), which are based on Halbert Dunn's seminal wellness concepts (Dunn, 1959). To the best of our knowledge, MULTIWD and WELLXPLAIN are the only datasets available for WDs. Task 1 (MULTIWD)

involves multi-label classification, while Task 2 (WELLXPLAIN) involves multi-class classification with expert annotator explanations, as summarized in Table 1. These datasets encompass six dimensions of wellness: Physical Aspect (PA), Intellectual Aspect (IA), Vocational Aspect (VA), Social Aspect (SA), Spiritual Aspect (SpA), and Emotional Aspect (EA). The definitions for these aspects by Sathvik and Garg (2023) can be found in § A.1.

**Task 1:** The MULTIWD dataset consists of 3281 instances, each comprising a text post and six distinct binary labels indicating whether a particular WD is present (1) or absent (0). The posts are crawled from Reddit's two most prominent mental health forums: r/Depression and r/SuicideWatch (Sathvik and Garg, 2023). Table 9 (§ A.5) presents the number of posts where a user explicitly refers to a mental health condition and specifies one or more wellness aspects impacted. In Table 9 (§ A.5), the users primarily mention depression, anxiety, and suicide as prominent MHCs impacting their social and emotional wellness. This corresponds to our intention of utilizing WD as a preliminary task to fine-tune LMs before engaging in binary mental health classification (refer to Figure 1).

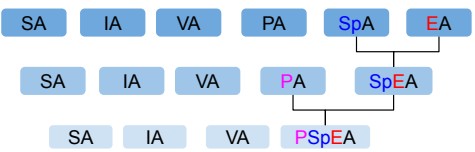

Figure 3: **Merging of WDs in MULTIWD.** The expert annotators suggest merging WDs based on their experience and literature (Bart et al., 2018).

**Subtask of Merging WDs in MULTIWD:** The WDs include many overlapping wellness tenets and individual proponents, making them unique. It is important to exercise the performance of LMs by merging WDs. The expert annotators suggest merging WDs based on their experience and literature (Bart et al., 2018). For instance, spiritual wellness is closely related to emotional wellness (in this specific clinical framework). Thus, we merge the most related classes in MULTIWD to explore how performance changes in an easier (4 classes) vs harder (5 to 6 classes) setting. Figure 3 shows the merging of WDs.

**Task 2:** The WELLXPLAIN dataset comprises 3092 instances from r/Depression and r/SuicideWatch. Each instance includes a text post,

accompanying explanatory information, and a specific WD aspect assignment. Table 9 (§ A.5) shows the presence of depression and anxiety as the top two MHCs expressed in the dataset impacting spiritual, emotional, social, and physical wellness. "Explanations" in WELLXPLAIN refer to the textual cues considered by annotators when determining the classification into one of the four predefined categories: (1) PA, (2) IVA, (3) SA, and (4) SPEA.

## 4 `WellDunn`: Methodology and Evaluation

**Domain-specific and General-purpose LMs:** We consider two distinct categories of models for the task of WD identification – general models and domain-specific models. The general models under consideration include BERT (Devlin et al., 2018), RoBERTa (Liu et al., 2019a), Xlnet (Yang et al., 2019), and ERNIE (Sun et al., 2019). Additionally, we incorporate domain-specific models, namely ClinicalBERT (Alsentzer et al., 2019), Mental-BERT (Ji et al., 2021), and PsychBERT (Vajre et al., 2021), to further explore their applicability within the mental health domain.

**Making LMs risk-averse with abstention:** To make the LM abstain from predictions when unconfident, we transform the model such that it makes a prediction only when certain (Liu et al., 2019b). WellDunn consists of classification tasks of the form $f : \mathbb{R}^{WXD} \to Y$, where BERT is used to generate textual encoding of a post with $W$ words. $Y$ represents the output of the classifier $f$, which can be one of the classes in the WELLXPLAIN and MULTIWD datasets. The LM responsible for classification is augmented with an abstention function $g : \mathbb{R}^{WXD} \to (0, 1)$, which is an extra sigmoid. Hence, LMs augmented with function $g$ learn using the Gambler's loss function (GL): $\mathcal{L}_{GL} = -\sum_{i=1}^{|Y|} y_i \log(\hat{y}_i + g)$, where $|Y|$ is the number of WDs in our case. In comparison to standard sigmoid cross entropy (SCE) loss, $\mathcal{L}_{GL}$ presents a confidence-oriented stricter bound on the performance of LMs, which is required for sensitive domains like mental health and well-being. This is because of a hyperparameter *Res*, which refers to *reservation*. The reservation is the fraction of the total test samples LMs predict and leave out $(1 - Res)$ uncertain samples.

**Large Language Models for WD Benchmarking:** We consider four LLMs in our benchmark-

**(a) SCE**

| Mo | 6-Labels | | 5-Labels | | 4-Labels | |
|---|---|---|---|---|---|---|
| | F1 | MCC | F1 | MCC | F1 | MCC |
| General models | | | | | | |
| B | **0.94** | **0.92** | 0.95 | 0.93 | 0.96 | 0.95 |
| R | **0.94** | **0.92** | **0.96** | 0.92 | 0.97 | 0.91 |
| E | 0.88 | 0.84 | 0.85 | 0.89 | **0.98** | 0.97 |
| X | 0.88 | 0.84 | **0.96** | **0.95** | 0.97 | 0.96 |
| Domain-specific models | | | | | | |
| P | 0.89 | 0.87 | 0.95 | 0.93 | **0.98** | 0.97 |
| C | 0.88 | 0.85 | **0.96** | **0.95** | 0.98 | **0.98** |
| M | 0.87 | 0.86 | 0.91 | 0.88 | 0.94 | 0.92 |

**(b) GL, 6-Labels**

| Mo | $Res=100\%$ | | $Res=95\%$ | | $Res=85\%$ | | $Res=75\%$ | |
|---|---|---|---|---|---|---|---|---|
| | F1 | MCC | F1 | MCC | F1 | MCC | F1 | MCC |
| General models | | | | | | | | |
| B | 0.64 | 0.55 | 0.63 | 0.55 | 0.62 | 0.54 | 0.60 | 0.54 |
| R | **0.71** | **0.63** | **0.71** | **0.63** | **0.70** | **0.62** | **0.69** | **0.61** |
| E | **0.71** | 0.62 | **0.71** | 0.62 | **0.70** | 0.61 | 0.68 | **0.61** |
| X | **0.71** | 0.62 | **0.71** | 0.62 | **0.70** | 0.61 | 0.68 | **0.61** |
| Domain-specific models | | | | | | | | |
| P | 0.65 | 0.55 | 0.64 | 0.55 | 0.63 | 0.54 | 0.62 | 0.54 |
| C | 0.62 | 0.51 | 0.62 | 0.51 | 0.62 | 0.51 | 0.61 | 0.52 |
| M | 0.68 | 0.59 | 0.67 | 0.59 | 0.66 | 0.58 | 0.65 | 0.58 |

**(c) GL, 5-Labels**

| Mo | $Res=100\%$ | | $Res=95\%$ | | $Res=85\%$ | | $Res=75\%$ | |
|---|---|---|---|---|---|---|---|---|
| | F1 | MCC | F1 | MCC | F1 | MCC | F1 | MCC |
| General models | | | | | | | | |
| B | 0.75 | 0.65 | 0.75 | 0.65 | 0.75 | 0.65 | 0.74 | 0.65 |
| R | **0.79** | **0.70** | **0.79** | **0.70** | **0.78** | **0.69** | **0.77** | **0.68** |
| E | **0.79** | 0.69 | 0.78 | 0.69 | **0.78** | **0.69** | **0.77** | **0.68** |
| X | 0.77 | 0.67 | 0.76 | 0.66 | 0.75 | 0.65 | 0.75 | 0.66 |
| Domain-specific models | | | | | | | | |
| P | 0.75 | 0.65 | 0.75 | 0.65 | 0.75 | 0.65 | 0.75 | 0.65 |
| C | 0.73 | 0.61 | 0.72 | 0.61 | 0.71 | 0.6 | 0.70 | 0.60 |
| M | 0.78 | 0.68 | 0.77 | 0.68 | 0.77 | 0.67 | 0.76 | 0.67 |

**(d) GL, 4-Labels**

| Mo | $Res=100\%$ | | $Res=95\%$ | | $Res=85\%$ | | $Res=75\%$ | |
|---|---|---|---|---|---|---|---|---|
| | F1 | MCC | F1 | MCC | F1 | MCC | F1 | MCC |
| General models | | | | | | | | |
| B | 0.82 | 0.72 | 0.81 | 0.72 | 0.81 | 0.72 | 0.81 | 0.72 |
| R | 0.83 | 0.72 | 0.82 | 0.72 | 0.82 | 0.72 | 0.81 | 0.72 |
| E | **0.84** | **0.75** | **0.84** | **0.75** | **0.84** | **0.75** | **0.84** | **0.75** |
| X | 0.83 | 0.73 | 0.83 | 0.73 | 0.82 | 0.73 | 0.83 | 0.73 |
| Domain-specific models | | | | | | | | |
| P | 0.81 | 0.71 | 0.81 | 0.71 | 0.81 | 0.71 | 0.81 | 0.71 |
| C | 0.77 | 0.66 | 0.77 | 0.66 | 0.77 | 0.66 | 0.76 | 0.65 |
| M | 0.83 | 0.72 | 0.82 | 0.72 | 0.82 | 0.72 | 0.82 | 0.72 |

Table 2: **Results on MULTIWD dataset**. (a) For Stochastic Cross-Entropy loss, merging labels from 6 to 4 significantly increases the accuracy. (b, c, d) Gambler's loss (GL) when predicting on 100% (0% abstention) of the data down to 75% (25% abstention). We see, as expected, that having fewer labels generally improves accuracy. Note that the GL does not perform effectively, abstaining from accurate and errant predictions at similar rates, resulting in a similar final accuracy. "Res" stands for "reservation."

ing: GPT-3.5, GPT-4, LLAMA, and MEDAL-PACA. GPT-4 is the latest in the GPT series and is considered state-of-the-art (OpenAI and et al., 2024). LLAMA is a recent LLM, similar to GPT-3.5, and MEDALPACA is a specialized version of LLAMA fine-tuned for medical data (Touvron et al., 2023; Han et al., 2023). Comparing MEDAL-PACA and LLAMA helps us understand the impact of fine-tuning on medical data, eliminating differences from the initial training of other LLMs. We utilize these LLMs in two strategies: (a) Prompting: We explore LLM performance on zero-shot (Kojima et al., 2022) and few-shot (Brown et al., 2020) prompting, and (b) Fine-tuning: We fine-tune LLAMA and MEDALPACA on the same data portion as the LMs as they are open-source. Figure 8 (§ A.5) provides the template for zero-shot prompting, which is later adapted for few-shot prompting by incorporating shots.

**Evaluation Strategy:** We utilize *SVD* on MULTIWD and WELLXPLAIN datasets to understand the complexity of the explanations produced for a prediction. Consider $M$ as the attention matrix of an LM. If we take the SVD of Matrix $M$, we will have the following: $M = USV$, where $U$ and $V$ are unitary arrays and $S$ is a vector with Singular Values (SVs). Considering the SVs, matrix $S$, we take the rank of this matrix and use it as the SVD

rank for every LMs used in this study. The lower the rank is for an LM, the lesser parts of the input the LM focuses on (Beren Millidge, 2023). Because clinical guidance on labeling the explanation is to select a concise and limited portion of the input as the determinant of a WD, the expected rank should be small to reflect that only a small portion of the input is needed. We compute the SVD rank for all LMs on both datasets.

We introduce an Attention-Overlap (AO) Score on WELLXPLAIN to assess if LMs focus on ground truth explanations. AO Score is calculated as the following: $AO = O/T$, where $O$ is the number of instances where the LM's estimated explanations overlap by at least 50% with corresponding WELLXPLAIN ground truth explanations, and $T$ is the total number of samples. The LM's estimated explanations are the top 4 tokens with the highest attention scores come from the attention matrix.

## 5 Experiments, Results, and Analysis

We employed the general architecture, depicted in Figure 2, consisting of two crucial steps applicable to four general and three domain-specific models. Step 1: We independently utilize each of the seven models to extract a representation for the input data. Step 2: This representation is fed into a fully connected neural network classifier, which determines

the aspect or dimension of the input.

**Experimental details**   Our experiments are categorized into two main groups: those involving Language Models (LMs) and those involving Large Language Models (LLMs). For the LMs, we fine-tune both general-purpose models (e.g., BERT, RoBERTa, XLNet, ERNIE) and domain-specific models (e.g., ClinicalBERT, MentalBERT, Psych-BERT) on two datasets: MultiWD and Wellxplain. These experiments utilize two loss functions: Softmax Cross Entropy (SCE) and Generalized Logit (GL). Performance is evaluated using metrics such as Precision, Recall, F1-score, Matthews Correlation Coefficient (MCC), and Accuracy for both datasets. Additionally, for Wellxplain, which provides ground truth explanations, we measure Attention Matrix Rank and Attention Overlap (AO) scores.

We utilize LLaMA, MedAlpaca, GPT-3.5, and GPT-4 in the LLM-related experiments. We fine-tune LLaMA and MedAlpaca, while GPT-3.5 and GPT-4 are prompted. These experiments follow a similar evaluation protocol to assess the models' performance across tasks. Table 8 (§ A) shows details of models employed in experimental Setup. Implementation details are also in § A.2.

**Research Question 1 (RQ1):**   *Does the performance of LMs depend on the number of WDs in the datasets, particularly in scenarios where experts define a hierarchical dependency between dimensions? Further, how do GL-trained LMs perform over SCE-trained?*  To answer this question, we conducted extensive experiments considering collapsing dimensions from six to four and evaluating the models using the F1 score to determine the relationship between decreasing the number of labels and model performance. Notably, general-purpose LMs perform significantly better than models fine-tuned to relevant social media and medical documents. Table 2 presents the results of employing general-purpose and domain-specific LMs, utilizing two different loss functions, namely SCE and GL, on the MULTIWD dataset.

All measurements improve from 6 to 4 dimensions, but the improvement rate varies between GL and SCE loss. This is observable under both the F1 and Matthews Correlation Coefficient (MCC) metrics in Table 2, where the GL improves at a higher rate (7%) as predictive classes are coalesced by the hierarchy compared to SCE (0.15). Our results indicate that improved predictive performance

can be obtained by focusing on lower-granularity labeling informed by clinical experts.

Table 2 shows that performance is not robust with respect to the loss function and can drop significantly. In the best case, the ERNIE model decreased by 6 points (from $85\%$ to $79\%$ ); in the worst case, the BERT model decreased by 34 points (from $94\%$ to $60\%$). Also note that GL assumes a desiderata: if the prediction is made with low confidence, the model should abstain from prediction because low-confidence data points are more likely to be predicted erroneously.

| | SCE | | GL | | | | | | | |
| | | | Res =100% | | Res =95% | | Res =85% | | Res =75% | |
| **Mo** | **F1** | **MCC** | **F1** | **MCC** | **F1** | **MCC** | **F1** | **MCC** | **F1** | **MCC** |
| | | | | General models | | | | | | |
| B | 0.80 | 0.79 | 0.82 | 0.79 | 0.81 | 0.78 | 0.74 | 0.73 | 0.62 | 0.60 |
| R | **0.82** | **0.80** | **0.84** | **0.81** | **0.83** | **0.80** | **0.77** | **0.76** | **0.64** | **0.62** |
| E | 0.67 | 0.76 | 0.83 | 0.80 | **0.83** | **0.80** | 0.76 | 0.74 | 0.63 | 0.61 |
| X | 0.77 | 0.78 | 0.82 | 0.80 | 0.82 | 0.79 | 0.74 | 0.73 | 0.63 | 0.60 |
| | | | | Domain-specific models | | | | | | |
| P | 0.79 | **0.80** | 0.78 | 0.75 | 0.77 | 0.74 | 0.70 | 0.69 | 0.60 | 0.58 |
| C | 0.78 | **0.80** | 0.71 | 0.69 | 0.70 | 0.68 | 0.62 | 0.62 | 0.52 | 0.51 |
| M | 0.80 | **0.80** | 0.82 | 0.79 | 0.81 | 0.79 | 0.73 | 0.73 | 0.62 | 0.60 |

Table 3: **Abstention Results on WELLXPLAIN**: Gambler's loss (GL) when predicting on 100% (0% abstention) of the data down to 75% (25% abstention). Where GL was only moderately ineffective in Table 2, it becomes actively harmful on WELLXPLAIN. We note the trend that for General models, the GL loss always results in the best performance, while SCE is best for Domain-specific models.

As shown in Table 2 and Table 3, we observe the opposite behavior in this data, where performance decreased by 2 points on average for MULTIWD and by 19 points on average for WELLXPLAIN as the reservation changed. One primary reason for this performance drop is the high abstention rates and the low number of samples in the dataset, which affect the number of predictions the model makes. Since GL introduces a "reservation" parameter, the model abstains from predicting when its confidence is low, reducing the total number of predictions and negatively impacting the final performance scores. We note that this may not be a generalizable observation about GL and more a function of our dataset and model types; however, it serves an important quantification that deep learning methods may not always transfer to medical applications and should be carefully validated before use.

**Research Question 2 (RQ2):**   *Given a ground-truth clinical explanation of the saliency of the input, do LMs learn to produce the same explanations (via attention maps) when producing a prediction?*  To answer this, we use SVD to compute the rank of the attention matrix to quantify the focus of LMs'

attention. We utilized the WELLXPLAIN, which incorporates ground truth explanations, to examine whether the models employ similar explanations to predict the outputs as experts.

| Mo | WELLXPLAIN | | MULTIWD | |
|---|---|---|---|---|
| | GL | SCE | GL | SCE |
| BERT | 31 | 54 | 53 | 52 |
| RoBERTa | 137 | 91 | 60 | 60 |
| Xlnet | 64 | 63 | 58 | 59 |
| ERNIE | 44 | 68 | **9** | **19** |
| ClinicalBERT | 35 | 50 | 39 | 40 |
| PsychBERT | 38 | 38 | 42 | 41 |
| MentalBERT | **30** | **30** | 47 | 45 |

Table 4: **Average attention matrix rank** via SVD for Gampler Loss (GL) vs Sigmoid Cross Entropy(SCE) across models where good explanations should have a lower rank. ERNIE achieves a **4.3** times lower rank on multi-label tasks. While MentalBERT had the best performance for WELLXPLAIN, it was not meaningfully better than other options. This shows that ERNIE is meaningfully better to use when explainable predictions are needed in multi-label environments. Note that we consider the average length of ground truth explanations as a good estimation of attention rank that is 25 on WELLX-PLAIN.

On average, LMs trained with GL show focused attention compared to SCE. This is desired in a critical application; we found better overlap while evaluating explanations coming from the attentions of LMs trained with GL versus SCE. Based on Table 4, we can see that SCE and GL usually explain similar complexity (i.e., similar rank). SCE and GL sometimes produce significantly higher-rank predictions and, in the RoBERTa case, produce nearly full-rank attention, indicating a lack of focus on individual portions of the input.

| MODELS | GL | | SCE | |
|---|---|---|---|---|
| | AO score | MCC | AO score | MCC |
| General models | | | | |
| BERT | 0.26 | 0.79 | 0.21 | 0.79 |
| RoBERTa | 0.05 | 0.81 | 0.00 | 0.80 |
| ERNIE | 0.28 | 0.80 | 0.26 | 0.76 |
| Xlnet | 0.23 | 0.80 | 0.03 | 0.78 |
| Domain-specific models | | | | |
| PsychBERT | 0.25 | 0.75 | 0.20 | 0.80 |
| ClinicalBERT | 0.13 | 0.69 | 0.10 | 0.80 |
| MentalBERT | 0.16 | 0.79 | 0.16 | 0.80 |
| Average | 0.19 | 0.78 | 0.14 | 0.79 |

Table 5: **Attention-overlap (AO) score for WELLXPLAIN**. It's noteworthy that even though various LMs achieve an MCC score exceeding 70%, their AO score barely surpasses 30%. For instance, RoBERTa exhibited an AO score of 0.0 despite an MCC score of 80%.MCCs come from Table 3.

This is further elucidated in Table 5, where we show the AO between the ground truth and the resulting attention from the model. In all cases, the AO is $\leq 28\%$. Notably, the general models have the highest AO (28%) compared to the domain-specific models (10-20%). This indicates a far more complex relationship between model

training matching the target distribution (domain-specific) and applicability to faithful downstream results (AO scores) than would be expected apriori.

In Figure 4, two posts are presented as input to the ERNIE model, which maintained consistent performance across all the experiments detailed in Table 2 and Table 3. In the first post, the model's accuracy in making predictions using SCE varied for different input dimensions. Specifically, it made incorrect predictions for 6-D and 5-D inputs while correctly predicting outcomes for 4-D inputs. Interestingly, the results differed when the model used the Gambler's Loss. It accurately predicted the outcomes for 6-D inputs, and the reservation value (0.0406) was low enough to support this prediction. However, for both 5-D and 4-D inputs, it made incorrect predictions by assigning two labels instead of the correct single label, which is also included in the prediction. The reservations (0.0676 and 0.0659) were relatively high compared to the 6-D case, which would call the model with GL to refrain from making the prediction.

Post 2 in Figure 4 shows how ERNIE with GL refrains from predicting because of relatively high reservation value compared to the ones mentioned in Post 1. The reservation values in GL don't vary significantly. Even a small decimal move can cause the LM not to make a prediction. In similar cases investigated, models with GL tend to refrain from predicting if the probability vector has fewer labels with nearly identical probabilities than the actual number of true labels (shown as ↓ in Figure 4). This characteristic of GL enables LMs to hold stringent confidence boundaries compared to using SCE.

**Research Question 3 (RQ3):** *How do LLMs perform when applied to the* WELLXPLAIN *dataset?* Our results have shown WELLXPLAIN to be more challenging than MULTIWD. As there is a growing interest in explainability in Language Models (LLMs), we focus on investigating LLMs using the WELLXPLAIN dataset. We investigate the performance of GPT-4 through prompting and apply fine-tuning over LLAMA and MEDALPACA. Zero-Shot GPT-4 scored 38% (MCC) lower than the best-performing LM (RoBERTa model, in Table 3) on WELLXPLAIN. This decline is attributed to GPT-4 lacking knowledge on the definitions and knowledge about wellness dimensions. To verify this finding, we applied few-shot prompting with five examples per class (20 total) to help GPT-4 recognize the pattern. Consequently, there was a

**Post 1:** I don't cry anymore. want to be around anyone do anything Work keeps me getting up everyday Without it would probably stare at my ceiling until passed back out again so tired know if there is a question in this There just isn't else tell.

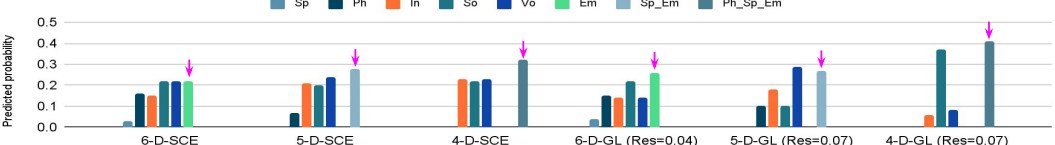

**Post 2:** I ve been on Viibryd for about a year give or take . The first few months gained 20 lbs and all together i've 25 ( last knew .) stopped weighing myself 2 ago because it was getting even more depressing 've read forums have noticed [...] - the won't bud ge same issue here 'm ready to ditch 1 sick of dealing going [...] can become harder treat over time tried probably 8 medications before found one really thought there hope beforehand [...] Does anyone currently take Viibryd ? Have you come off like heard.

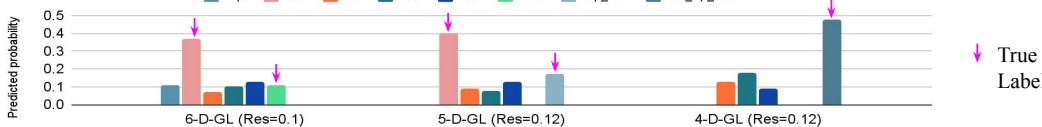

Figure 4: Bar plots illustrating the predicted probabilities from ERNIE LM fine-tuned on MULTIWD. These outcomes offer a visual perspective on the two posts, revealing the contrast between GL and SCE across the 6, 5, and 4 dimensions (D). Notably, in the case of post 2, the ERNIE model with GL abstains from making the prediction. Note that the highlighted posts are obtained from SCE with 4-D. The highlighted posts for GL with 4-D and more posts are in Figure 6 and Figure 7 (§ A.5).

10% improvement in performance. We fine-tune LLaMA and MEDALPACA since they are open source (refer to Table 6). Although there was an improvement compared to GPT-4, the performance gain is not substantial, mainly because of the limited size of the WellDunn.

| LLM | A | P | R | F1 | MCC | AO | AR |
|-----|-----|-----|-----|-----|-----|-----|-----|
| LLAMA | **0.73** | 0.73 | **0.71** | **0.65** | 0.56 | 0 | 63 |
| MEDALPACA | 0.68 | 0.73 | 0.69 | 0.63 | **0.59** | 0 | **21** |

Table 6: MEDALPACA surprisingly performs worse on WELLXPLAIN task despite being a fine-tuned LLAMA on medical data. This shows how fine-tuning the domain is not a guarantee of increased performance. Therefore, thorough validation in medical contexts is necessary. A: Accuracy, P: Precision, R: Recall, AO: Attention Overlap score, AR: Attention Rank via SVD.

**Error Analysis**: We conducted a detailed analysis of attention maps for LMs trained using SCE and GL. *Low correlation between attention and performance*: Despite the fact that SCE has a higher performance than GL (when at least 15% abstention) shown in Table 3, GL has higher AO scores than SCE (see Table 5 and Figure 7 (§ A.5) for further details). The fact that this misalignment does not improve even as models increase in accuracy suggests that the models might be "right for the wrong reasons," potentially leveraging spurious correlations or biases present in the training data rather than genuinely understanding underlying clinical concepts.

*Imperfect explanation:* One might argue that an imperfect explanation is acceptable when perfor-

mance metrics are high. However, in mental health, a prediction without a proper explanation is insufficient. Given the potential for models to be "right for the wrong reasons," it becomes essential to incorporate a more relevant, domain-specific context when preparing models for mental health tasks. To address this, a human-AI teaming approach, where experts provide explicit feedback, could prove invaluable. We suggest exploring this strategy in future research.

**Research Question 4 (RQ4):** *Are larger models Panacea for NLP applications in Mental Health?* One may wonder if still larger LMs, like GPT-3.5 and GPT-4, would perform better and resolve the issues we observe. Though we can not inspect the attention scores of these proprietary models, their relative performance can give us some insights as to how this mildly out-of-distribution data (it is all English, but not typical text) nature impacts performance. Apriori, one might expect high performance on WELLXPLAIN given their exposure to various healthcare datasets up to 2023, The WELLXPLAIN dataset presents two unique challenges: (1) It is not focused on predicting mental health conditions, as is common with earlier datasets. Instead, these models must identify relevant aspects of declining wellness to generate appropriate Wellness Definitions (WDs). (2) The WDs are based on Halbert Dunn's well-known definition, likely familiar to the models.

Table 7 shows that when evaluated using the

| Model | Accuracy | Precision | Recall | F1 | MCC |
|---|---|---|---|---|---|
| GPT-4 (Zero-shot) | 0.53 | 0.69 | 0.53 | 0.53 | 0.43 |
| **GPT-4 (5-shot)** | **0.63** | **0.75** | **0.63** | **0.64** | **0.54** |
| GPT-4 (10-shots) | 0.59 | 0.68 | 0.59 | 0.60 | 0.48 |
| GPT-4 (15-shot) | 0.57 | 0.77 | 0.57 | 0.58 | 0.50 |
| GPT-4 (20-shot) | 0.58 | 0.75 | 0.57 | 0.59 | 0.49 |
| GPT-4 (40-shot) | 0.49 | 0.70 | 0.51 | 0.49 | 0.41 |
| GPT-3.5 (Zero-shot) | 0.38 | 0.68 | 0.43 | 0.39 | 0.34 |
| GPT-3.5 (5-shot) | 0.38 | 0.67 | 0.43 | 0.39 | 0.34 |
| GPT-3.5 (10-shot) | 0.38 | 0.68 | 0.43 | 0.39 | 0.34 |
| GPT-3.5 (15-shot) | 0.37 | 0.67 | 0.42 | 0.38 | 0.30 |
| GPT-3.5 (20-shot) | 0.38 | 0.68 | 0.42 | 0.38 | 0.30 |
| GPT-3.5 (40-shot) | 0.36 | 0.67 | 0.41 | 0.36 | 0.28 |

Table 7: **Performance of GPT-4 and GPT-3.5 on Zero-Shot and Few-Shot** prompting. Providing 5 examples per class (FEW-SHOT₅),GPT-4's performance boosts by 10, 4, 6, 5, 14 points compared to ZERO-SHOT, 10, 15, 20, 40 shots, respectively. The same prompt was used for both GPT-4 and GPT-3.5. 400 samples from WELLXPLAIN dataset were selected randomly as test data for these experiments.

robust Matthews Correlation Coefficient (MCC), both GPT-3.5 and GPT-4 underperformed, showing minimal or negligible improvement in probability scores. Few-shot prompting did not meaningfully improve results. This result highlights the importance of smaller, local models and the need to validate the explanation's alignment with a ground-truth physician practice, as canonical NLP assumptions don't always apply to this data.

## 6 Conclusion and Future Work

`WellDunn` introduces a demanding pair of datasets for the AI for Social Impact community working on mental health. Through thorough benchmarking on domain-specific and general-purpose LMs, we've highlighted the disparities between prediction accuracy and attention, underscoring the need for a transparent classifier rooted in clinical understanding. Second, despite the expectation that Gambler's Loss would enhance performance by avoiding predictions for low-confidence samples, we observed a significant drop in performance for the MULTIWD dataset. Third, the AO scores show that attention explanations are not closely aligned with the ground truth. Further experiments were conducted to thoroughly analyze the datasets and confirm these findings refer to Table 10-Table 16 (§ A.5). Finally, we extended our investigation to LLMs such as GPT-4, LLAMA, and MEDALPACA through prompting and fine-tuning. Surprisingly, LLMs underperformed. Despite this, there is still potential for experimenting with different prompting and retrieval-augmented generation (RAG) strategies. While retrieval-augmented methods like RAG can enhance LLM performance, they add complexity and require extensive knowledge curation and developing a suitable dataset for mental health, which we leave for future work (for more, see § A.4). A complete GitHub repository containing our code is provided (see § A.3).

## Acknowledgements and Funding Disclosure

We would like to extend our heartfelt thanks to **Dr. Muskan Garg** for sharing unique datasets that offer valuable insights into mental health and well-being. We also greatly appreciate her insightful suggestions on the manuscript. We would also like to thank the anonymous reviewers for their valuable comments, questions and suggestions.

This material is based in part upon work supported by the National Science Foundation under Grant No. IIS-2024878 and the Army Research Laboratory, Grant No. W911NF2120076. The U.S. Government is authorized to reproduce and distribute reprints for Governmental purposes notwithstanding any copyright notation thereon. The views and conclusions contained herein are those of the authors and should not be interpreted as necessarily representing the official policies or endorsements, either express or implied, of the U.S. Government.

## Ethical Considerations

In this study, we utilize publicly available datasets, MULTIWD and WELLXPLAIN, both of which have been carefully designed to prioritize user privacy and mitigate data exposure risks. The MULTIWD and WELLXPLAIN datasets, which consist of anonymized posts, labels, and expert-provided explanations (in the case of WELLXPLAIN), do not include any associated metadata that could reveal personal identifiers, such as names or demographic attributes. These datasets underwent a rigorous de-identification process prior to their use, ensuring that only anonymized content was included. Posts were fragmented into sentences and short paragraphs, labeled independently, and randomly shuffled to further reduce re-identification risks. The datasets do not contain usernames (or anonymous Reddit IDs) or identifying fields that can pose a risk to user identification. All the posts have been anonymized, obfuscated, and rephrased to avoid linking data across sites (Sathvik and Garg, 2023), consequently preventing potential privacy breaches. For more details about the ethical considerations regarding to the datasets, we refer to (Sathvik and Garg, 2023; Garg, 2024; Liyanage et al., 2023).

## Limitations

While `WellDunn` is the first attempt to assess finer aspects of wellness influencing mental health conditions, there are limitations in the benchmark's completeness. The dataset allows for a thorough examination of language models in identifying wellness determinants and providing explanations. However, inconsistencies in attention and explanation levels exist, especially in models trained for specific domains compared to general-purpose models, including LLMs. This raises concerns about the consistency and reliability of predictions and generated explanations, posing an open challenge for LLMs (Gaur and Sheth, 2024). We leave this challenge as an open avenue for future work to address.

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

# A Appendix

## A.1 Wellness Dimension (or Aspect) Definitions

- **Physical Aspect (PA)**: Physical wellness fosters healthy dietary practices while discouraging harmful behaviors like tobacco use, drug misuse, and excessive alcohol consumption. Achieving optimal physical wellness involves regular physical activity, sufficient sleep, vitality, enthusiasm, and beneficial eating habits. Body shaming can negatively affect physical well-being by increasing awareness of medical history and appearance issues.

- **Intellectual Aspect (IA)**: Utilizing intellectual and cultural activities, both inside and outside the classroom, and leveraging human and learning resources enhance the wellness of an individual by nurturing intellectual growth and stimulation.

- **Vocational Aspect (VA)**: The Vocational Dimension acknowledges the role of personal gratification and enrichment derived from one's occupation in shaping life satisfaction. It influences an individual's perspective on creative problem-solving, professional development, and the management of financial obligations.

- **Social Aspect (SA)**: The Social Dimension highlights the interplay between society and the natural environment, increasing individuals' awareness of their role in society and their impact on ecosystems. Social bonds enhance interpersonal traits, enabling a better understanding and appreciation of cultural influences.

- **Spiritual Aspect (SpA)**: The Spiritual Dimension involves seeking the meaning and purpose of human life, appreciating its vastness and natural forces, and achieving harmony within oneself.

- **Emotional Aspect (EA)**: The Emotional Dimension enhances self-awareness and positivity, promoting better emotional control, realistic self-appraisal, independence, and effective stress management.

## A.2 Implementation Details

We utilized the Ada GPU cluster for our implementation, leveraging RTX 6000 and RTX 8000 GPUs. The cluster comprises 13 nodes equipped with two 24-core Intel Cascade Lake CPUs and varying GPU configurations, providing a robust computing environment with high-performance capabilities.

In our experiments, we employed a common data partitioning strategy, splitting each dataset into an 80% training set and a 20% test/validation set. This division allows us to train our models on a substantial portion of the data while evaluating their performances on an independent subset, ensuring a robust assessment of their generalization capabilities.

In our implementation for LLMs, we utilized the GPT-4 model, specifically the *gpt-4-0613* version. This version is a snapshot of GPT-4 from

June 13th, 2023 and has a context window of 8192 tokens. It is trained on data up to September 2021. We also assesss the performance of ChatGPT using GPT-3.5 turbo (GPT-3.5-TURBO version). Additionally, for the LLAMA model, we used *orca_mini_3b*[5] with 3 billion parameters, which is an OpenLLaMa-3B model, trained on explain tuned datasets, created using Instructions and Input from WizardLM, Alpaca and Dolly-V2 dataset. Another LLama model that we employed in our experiments, is MEDALPACA-7B[6] which is based on LLaMA (Large Language Model Meta AI) and contains 7 billion parameters. More implementation details of LMs and LLMs used in our work are shown in Table 8 (§ A).

For our LLMs experiments, it costs $ 131 for GPT-4 usage. In addition, we used Colab Pro from Google, which costs $ 105.98.

Additional implementation details are available in the code associated with each model. Due to space constraints, only the details for fine-tuning the LLama model are presented, as shown in Figure 5.

### A.3 Reproduciblity

Our WellDunn framework is straightforward to implement and easily reproducible. We have included the source code and data, along with a comprehensive README file containing detailed instructions on how to run the code. A GitHub link to access the code is provided:

- https://github.com/vedantpalit/WellDunn

### A.4 Broader Considerations

1. **How can WellDunn serve as a solution to immediate potential problems concerning social impact?** WellDunn is a response to a critical need in the landscape of LMs applied to mental health analysis. As observed in forums like CLPsych, the current trend primarily revolves around creating crowdsourced datasets. However, these lack the robust theoretical and empirical frameworks crucially employed by mental health professionals, volunteers, and counselors. Consequently, LMs' true utility and effectiveness in this context remain inadequately assessed and accepted.

Our benchmark, WellDunn, aims to bridge this gap by complementing existing initiatives that leverage LMs for understanding textual conversations around mental health. As highlighted by Gross et al. (2019), mental health issues often stem from deteriorating mental well-being. WellDunn's unique approach involves compelling LMs to identify causal cues of mental illness, align them with concept classes from Dunn's framework, and, importantly, elucidate the rationale behind selecting these specific causal cues. This structured approach intends to enhance the depth and accuracy of LMs in comprehending and addressing mental health concerns.

2. **Are there any implementation issues when WellDunn is considered for in practice?** Through rigorous benchmarking, it became evident that the ERNIE LM (better performance, considering different dimensions, SCE, and GL, among other models) exhibits significant potential for responsible and effective performance. Its demonstrated attributes include commendable accuracy, concentrated attention, and enhanced explanatory capabilities. These findings strongly indicate the feasibility of fine-tuning this model for subsequent applications within the mental health domain. Since the model and our dataset will be publicly available with proper code and implementation details, we don't foresee any issue concerning reproducibility.

3. **Is the dataset realistic?** The dataset for WellDunn is meticulously designed using Dunn's Wellness Index as its foundation. This established index, developed by Dr. Halbert L. Dunn in the 1960s, is widely recognized and employed in various fields, including health education, nursing, and public health (Dunn, 1959; Logan et al., 2023; Liyanage et al., 2023). Dunn's framework conceptualizes well-being not merely as the absence of disease but rather as a dynamic process of growth and self-actualization. By leveraging Dunn's Wellness Index as its foundation, the WellDunn dataset offers several advantages:

   - Validity and Reliability: Dunn's framework is well-validated and has shown consistent results in numerous research studies. This ensures the dataset's relia-

---

[5]https://huggingface.co/pankajmathur/orca_mini_3b
[6]https://huggingface.co/medalpaca/medalpaca-7b

| | Model | Version, # parameters | Link | GL/SCE_BS | MAx-Len | training rate |
|---|---|---|---|---|---|---|
| 1 | BERT | bert-base-uncased, 110M | https://huggingface.co/google-bert/bert-base-uncased | 32 | 64 | 0.00001 |
| 2 | Roberta | roberta-base, 125M | https://huggingface.co/FacebookAI/roberta-base | 32 | 64 | 0.00001 |
| 3 | XLNET | xlnet-base-cased, 110M | https://huggingface.co/xlnet/xlnet-base-cased | 2 | 64 | 0.00001 |
| 4 | ERNIE | ernie-2.0-base-en, 110M | https://huggingface.co/nghuyong/ernie-2.0-base-en | 1 | 256 | 0.00001 |
| 5 | ClinicalBERT | Bio_ClinicalBERT, - | https://huggingface.co/emilyalsentzer/Bio_ClinicalBERT | 1 | 256 | 0.00001 |
| 6 | PsychBERT | psychbert-cased, - | https://huggingface.co/mnaylor/psychbert-cased | 32 | 64 | 0.00001 |
| 7 | MentalBERT | mental-bert-base-uncased, - | https://huggingface.co/mental/mental-bert-base-uncased | 2 | 64 | 0.00001 |
| 8 | LLaMa | orca_mini_3b, 3B | https://huggingface.co/pankajmathur/orca_mini_3b | 2 | 64 | 0.001 |
| 9 | Medalpaca | medalpaca-7b, 7B | https://huggingface.co/medalpaca/medalpaca-7b | 2 | 64 | 0.001 |
| 10 | GPT-3.5 | gpt-3.5-turbo | - | - | - | - |
| 11 | GPT-4 | gpt-4-0613 | - | - | - | - |

Table 8: **Details of Models Employed in Experimental Setup** For each experiment/model, we utilized three different random states: 200, 345, and 546. Each model was trained for 5 epochs. GL/SCE_BS stands for GL or SCE Batch Size. Note that the batch size should be set appropriately in our code provided in the GitHub link (§ A.3) based on this table.

bility and accuracy in measuring mental well-being.

- Holistic Perspective: Dunn's comprehensive framework captures the multidimensional nature of mental well-being, encompassing physical, social, emotional, intellectual, and spiritual dimensions. This holistic approach provides a richer understanding of mental health than focusing solely on symptoms or diagnoses.

As a future effort, the `WellDunn` dataset's connection to Dunn's framework allows for tailored interventions based on individual needs and strengths across different dimensions of well-being. This personalized approach leads to more effective and sustainable improvements in mental health.

4. **How much can identifying wellness indicators in mental health research contribute to enhancing clinical outcomes?** Research at the juncture of mental health and AI, often driven by a collaboration between clinical psychologists, linguists, and AI researchers, has primarily focused on classifying textual expressions into identifiable mental health disorders. Yet, it's pivotal to recognize that MHCs stem from various causal events impacting an individual's well-being, ranging from personal crises like divorce or academic struggles to societal issues like gender bias. Understanding these causal cues holds immense significance alongside identifying emerging MHCs. It's about detecting the disorder and unraveling the underlying triggers. Equally crucial is the ability to provide clear and comprehensive explanations to aid comprehension, a vital aspect often overlooked in current models. However, integrating these dual tasks –

causality detection and explanatory capabilities – within existing LMs presents multifaceted challenges. Our extensive benchmarking efforts form the foundation underscoring the complexity of addressing the e challenges.

5. **It is not only dimension but also the degree of mental illness that clinicians identify. How would this issue be addressed?** It is crucial for clinicians to identify the dimension and degree of mental illness for effective diagnosis and treatment. `WellDunn` addresses this issue by specifically focusing on WD cues, which play a significant role in the development and progression of mental illness. Prolonged neglect of these WD factors, including comorbid conditions, can exacerbate the severity of mental illness, making it more challenging to manage. `WellDunn` tackles this problem by providing annotated data that links specific causal factors to the associated MHCs. Additionally, the dataset includes multiple instances with extracted wellness-specific cues, enabling researchers and clinicians to analyze the impact of various factors on mental health outcomes. This comprehensive approach allows for more nuanced and accurate mental health assessments, ultimately leading to improved diagnosis, treatment, and prevention strategies.

6. **Is attention the only mechanism to identify what the model focuses on?** An effective and usable approach for identifying what a model focuses on depends on several factors, including the specific model architecture, the task at hand, and the desired level of interpretability. Attention offers a good initial glimpse into the model's focus, but combining it with other techniques is often valuable for

a more comprehensive understanding. In our benchmarking process, we examine attention in the following three different ways:

(a) Self-attention, a low-rank mechanism in LMs, serves to elucidate the models' comprehension of entities and associations within data. By performing SVD on the attention matrix of transformer models and mapping them to token embeddings, discernible semantic clusters emerge. The rank of this attention matrix significantly impacts the model's capacity to capture and represent diverse data relationships. A higher rank signifies a broader representation of relationships, possibly indicating a lack of specificity or inadequate contextual understanding in the model's input text interpretation. Conversely, a lower rank is crucial for the model's effectiveness in tasks emphasizing nuanced language comprehension. Nonetheless, some argue that the SVD rank might not entirely define the model's expressiveness.

(b) Attention Maps over Text: We explored the visualization of attention weights across tokens in a sequence. This approach, known as the Attention Maps, visually explains which set of tokens contribute to prediction and which set of tokens were overlooked. Even in this method, someone might argue that attention maps can be noisy and misleading, highlighting specific tokens but failing to capture the broader context and interactions between them.

(c) Attention Overlap Scoring (AO Score): AO Scoring emphasizes the importance of aligning LMs with domain-specific knowledge. By leveraging explanations provided by experts in the field, this method assesses how accurately and effectively LMs focus on the relevant parts of the input data. For instance, in medical or legal domains where specific terms or concepts hold paramount importance, this approach ensures that the model's attention aligns with what experts in those fields deem crucial.

Other techniques, like Layerwise Relevance Propagation (Montavon et al., 2019), Atten-

tion Visualization (Vig, 2019), and LIME (Ribeiro et al., 2016)) offer alternative avenues for explainability. However, the interpretation derived from these methods aligns with the findings presented here.

7. **Is there a limitation of this study because of the data source and availability, and if this could be carried out in big data terms, would it reproduce similar results and insights?** We anticipate consistent results when applying our methodology to different mental health topics. We have confidence in the model's predictive capabilities and ability to focus on salient aspects of the prediction task, making it adaptable to various mental health domains without significant deviations in outcomes.

8. **This research is based on a few mental health topics. To what extent would this work produce different insights if applied to different mental health topics?** While the WellDunn benchmark currently focuses on depression, anxiety, bipolar, schizophrenia, and suicide risk, its underlying framework and methodology have the potential to be applied to a variety of other mental health topics.

(a) MHCs vary in presentation and underlying mechanisms, but the causal factors and wellness dimensions intersect. For instance, poor physical health can negatively impact mental well-being and vice versa. Similarly, social isolation can affect emotional well-being, and spiritual well-being can influence how individuals cope with stress. Also, a lack of physical activity contributes to depression.

(b) The effectiveness of LMs in detecting causal cues might vary across conditions – We have identified such a phenomenon but did not explicitly discuss it.

(c) The ethical considerations and potential risks might differ depending on the mental health condition. Applying the Well-Dunn framework to conditions with higher stigma or vulnerability, such as personality disorders or eating disorders, might require additional safeguards and ethical considerations.

9. **How do we apply the results of the cur-**

rent study with other datasets? **Considering that the majority of prior research on mental health information focused on multimodal information.** Most of the previous datasets in mental health focus on text, with only a few including multiple modalities of information. This is mainly because people are worried about social judgment and keeping their information private. So, we've concentrated on text-only datasets in mental health. We aim to complement these efforts by creating a benchmark that helps accurately identify MHCs and explain the results. Yazdavar et al. (2020) discusses mental health using different types of information, like images or videos, and we want to build on that. We will adjust their dataset by adding explanations and labeling other details related to well-being. As for the text part, we will use the best model we found through our benchmarking (ERNIE) to improve our understanding of mental health through text.

10. **What are possible explanations for this? LLMs show lower performance when compared to the highest-performing fine-tuned LMs.**

This is a counterintuitive finding since recent research indicated that MedAlpaca sometimes surpassed fine-tuned language models, particularly with multi-stage pre-training and alignment strategies. However, we still agree with our findings that these models (including LLMs) can be right for the wrong reasons, which can be dangerous for mental health. Prior studies on Gallatica and MedAlpaca did not investigate the aspects of attention and explanation in LLMs (He et al., 2023). We believe it is crucial to conduct further experiments on large language models (LLMs) in mental health, emphasizing the need for datasets that include expert explanations.

11. **Why did we not use Chain-of-Thought (CoT)?**

Techniques like Singular Value Decomposition (SVD) and attention overlap score are particularly useful for directly analyzing and quantifying the relationships between attention mechanisms and ground truth explanations. In our case, the ground truth explanations are not human-like but rather specific parts of the textual post. Therefore, CoT, which excels in generating detailed, human-like reasoning, does not add significant value in this context. SVD and attention overlap score align more with our task requirements, providing a clear and efficient evaluation of the model's performance. More details in (Chen et al., 2024; Han et al., 2022) :

12. **Why did we not approach `WellDunn` as a named-entity recognition (NER) task to find evidence could significantly improve the AO results?**

(a) Unlike clear-cut entities like names or locations, descriptions of wellness issues can be vague, subjective, and can vary significantly. (b) Context is required: For example, the statement "I'm feeling blue" could be a colloquial way of expressing sadness, or it could be a clinical indication of depression, depending on additional contextual information. (c) Mood swings, anxiety, and sleep disturbance can affect different dimensions of wellness. A NER system would need to disambiguate such terms within specific contexts, a task that can be particularly complex without additional information or specialized knowledge requirements, such as an ontology for the wellness dimension.

## A.5 Extra figures and tables for more detailed information

In this section, we provided more detailed information regarding our results. Figure 6 shows two highlighted posts for GL with 4-D. In addition, Figure 7 provides sample posts that are classified correctly using ERNIE model using SCE but incorrectly with GL. Moreover, Table 10 to Table 16, provide more details of our experimental results.

| MHC | MULTIWD | | | | | | WELLXPLAIN | | | |
|---|---|---|---|---|---|---|---|---|---|---|
| | SpA | PA | IA | SA | VA | EA | PA | IVA | SA | SpEA |
| Depression | 40 | 292 | 159 | 519 | 148 | 425 | 68 | 22 | 61 | 27 |
| Bipolar | 0 | 13 | 5 | 14 | 7 | 15 | 6 | 1 | 1 | 1 |
| Anxiety | 9 | 132 | 77 | 181 | 56 | 210 | 35 | 11 | 27 | 31 |
| Schizophrenia | 1 | 4 | 1 | 3 | 2 | 1 | 1 | 0 | 1 | 0 |
| Suicide | 8 | 63 | 39 | 160 | 30 | 124 | 9 | 5 | 7 | 8 |

Table 9: **Distribution of Mental health conditions (MHCs) in MULTIWD and WELLXPLAIN:** Number of posts explicitly mentioning an MHC and specifying affected wellness aspects.

```
(model): LlamaModel(
  (embed_tokens): Embedding(32001, 4096, padding_idx=32000)
  (layers): ModuleList(
    (0-31): 32 x LlamaDecoderLayer(
      (self_attn): LlamaAttention(
        (q_proj): Linear(in_features=4096, out_features=4096, bias=False)
        (k_proj): Linear(in_features=4096, out_features=4096, bias=False)
        (v_proj): Linear(in_features=4096, out_features=4096, bias=False)
        (o_proj): Linear(in_features=4096, out_features=4096, bias=False)
        (rotary_emb): LlamaRotaryEmbedding()
      )
      (mlp): LlamaMLP(
        (gate_proj): Linear(in_features=4096, out_features=11008, bias=False)
        (up_proj): Linear(in_features=4096, out_features=11008, bias=False)
        (down_proj): Linear(in_features=11008, out_features=4096, bias=False)
        (act_fn): SiLUActivation()
      )
      (input_layernorm): LlamaRMSNorm()
      (post_attention_layernorm): LlamaRMSNorm()
    )
  )
  (norm): LlamaRMSNorm()
)
(dropout): Dropout(p=0.3, inplace=False)
(linear): Linear(in_features=4096, out_features=4, bias=True)
)
```

Figure 5: **Implementation details:** Structure of LLama model used for fine-tuning.

**Post 1:** I don't cry anymore. want to be around anyone, do anything. Work keeps me getting up every day. Without it would probably stare at my ceiling until passed back out again m so tired know if there is a question in this, There just isn else tell.

**Post 2:** I ve been on Viibryd for about a year, give or take. The first few months gained 20 lbs, and all together, I've 25 ( last knew .) stopped weighing myself 2 ago because [...] weight gain with this medication, no matter how hard they worked - they won't budge; same issue here I'm ready to ditch 1 sick of dealing going gym 3 6 times week has done nothing seems to increase my anxiety at Its expensive coupon code insurance thing is VERY scared as know depression can become harder treat over time tried probably 8 medications before found one really thought there hope beforehand stuck out obviously mental health important but causing me to obsess what eat often workout Does anyone currently Viibryd? Have you come off like heard.

Figure 6: The highlighted posts 1 and 2 were obtained from RoBERTa with GL with 4-D. The results show that RoBERTa's fine-tuning using GL makes its attention more focused compared to SE. For instance, "depression" and "Viibryd" are highlighted and captured by Roberta when tuned with GL as opposed to SCE. Note that this example should be read in Figure 4. The figure shows the attention map of RoBERTa fine-tuned with SCE.

| # | Highlighted output posts for SCE and GL based loss functions |
|---|---|
| 1 | **SCE**: If someone can give me a link that would be nice . I might fit in more or i talk to other people , don 't know. 
 **GL**: If someone can give me a link that would be nice . I might fit in more or i talk to other people , don't know. |
| 2 | **SCE**: I have decided to do myself a favour and clean my room . These past years not been very good me Towards the beginning of month moved into this My best friend hasn't talked since out from where she lives shrink in 2. 
 **GL**: I have decided to do myself a favour and clean my room . These past years not been very good me Towards the beginning of month moved into this My best friend hasn't talked since out from where she lives shrink in 2 . |
| 3 | **SCE**: I have been diagnosed with anxiety and depression right now taking prescription med for the last couple weeks . It really helps ! A little background - got out of a 6 year relationship due to not seeing future my ex in December 2019 And then one person who thought was. 
 **GL**: I have been diagnosed with anxiety and depression right now taking prescription med for the last couple weeks . It really helps ! A little background - got out of a 6 year relationship due to not seeing future my ex in December 2019 And then one person who thought was. |
| 4 | **SCE**: My mom had a talk with me about how if it wasn't for she would give up . Now suicide is off the table , but what fuck on then ? Living through this hell where i cant concentrate because have intrusive thoughts so bad NEED something to take my mind. 
 **GL**: My mom had a talk with me about how if it wasn't for she would give up . Now suicide is off the table , but what fuck on then ? Living through this hell where i cant concentrate because have intrusive thoughts so bad NEED something to take my mind. |
| 5 | **SCE**: I can 't take this anymore . 've been wanting to buy a pocket pistol or similar weapon off myself with for the past few days now , and doing research live in SE Michigan drive money ( afford living ), have studying get certification. 
 **GL**: I can 't take this anymore . 've been wanting to buy a pocket pistol or similar weapon off myself with for the past few days now , and doing research live in SE Michigan drive money ( afford living ), have studying get certification. |

Figure 7: The highlighted outputs of SCE and GL-based loss function for five different input posts where the RoBERTa model classified the input correctly using SCE but incorrectly using GL. Note that the shiner blue color has a higher score of attention.

| Model | #label(#samples) | Recall | Precision | F-Measure | MCC | Accuracy |
|---|---|---|---|---|---|---|
| ERNIE | | 0.88 | 0.89 | 0.88 | 0.84 | 0.87 |
| XLNET | | 0.86 | 0.91 | 0.88 | 0.84 | 0.88 |
| PsychBERT | | 0.88 | 0.9 | 0.89 | 0.87 | 0.85 |
| ClinicalBERT | 6 (1186) | 0.87 | 0.92 | 0.88 | 0.85 | 0.88 |
| MentalBERT | | 0.86 | 0.9 | 0.87 | 0.86 | 0.86 |
| BERT | | **0.95** | **0.95** | **0.95** | **0.93** | **0.94** |
| RoBERTa | | 0.93 | **0.95** | 0.94 | 0.92 | 0.93 |
| ERNIE | | 0.85 | 0.88 | 0.85 | 0.89 | 0.83 |
| XLNET | | 0.95 | 0.96 | **0.96** | **0.95** | 0.94 |
| PsychBERT | | 0.95 | 0.95 | 0.95 | 0.93 | 0.94 |
| ClinicalBERT | 5 (1172) | **0.96** | **0.97** | **0.96** | **0.95** | **0.95** |
| MentalBERT | | 0.91 | 0.92 | 0.91 | 0.88 | 0.94 |
| BERT | | 0.94 | 0.95 | 0.94 | 0.87 | 0.91 |
| RoBERTa | | 0.95 | 0.96 | 0.96 | 0.92 | 0.94 |
| ERNIE | | **0.98** | **0.98** | **0.98** | 0.97 | **0.98** |
| XLNET | | 0.97 | 0.97 | 0.97 | 0.96 | 0.97 |
| PsychBERT | | **0.98** | 0.97 | **0.98** | 0.97 | 0.97 |
| ClinicalBERT | 4 (1104) | **0.98** | **0.98** | **0.98** | **0.98** | 0.96 |
| MentalBERT | | 0.95 | 0.95 | 0.94 | 0.92 | 0.95 |
| BERT | | 0.95 | 0.95 | 0.94 | 0.91 | 0.94 |
| RoBERTa | | 0.97 | 0.97 | 0.97 | 0.91 | 0.96 |
| ERNIE | | 0.95 | 0.94 | 0.96 | 0.94 | 0.95 |
| XLNET | | 0.94 | 0.94 | 0.95 | 0.95 | 0.94 |
| PsychBERT | | 0.94 | 0.94 | 0.95 | 0.95 | 0.95 |
| ClinicalBERT | 3 (1072) | 0.95 | 0.95 | 0.95 | 0.95 | 0.97 |
| MentalBERT | | 0.97 | 0.97 | 0.97 | 0.94 | 0.96 |
| BERT | | 0.98 | 0.96 | 0.97 | 0.96 | 0.96 |
| RoBERTa | | **0.99** | **0.98** | **0.99** | 0.97 | **0.98** |

Table 10: Performance of models on **MULTIWD** dataset for **SCE** across various dimensionalities.

| Model | Precision | Recall | F-Measure | Support | MCC | Accuracy |
|---|---|---|---|---|---|---|
| ERNIE | 0.78 | 0.70 | 0.67 | 618 | 0.76 | 0.82 |
| XLNET | 0.82 | 0.81 | 0.77 | 618 | 0.78 | 0.81 |
| PsychBERT | 0.82 | **0.84** | 0.79 | 618 | **0.80** | 0.82 |
| MentalBERT | 0.83 | **0.84** | 0.80 | 618 | 0.80 | 0.83 |
| BERT | 0.86 | 0.82 | 0.80 | 618 | 0.79 | 0.84 |
| RoBERTa | **0.87** | 0.83 | **0.82** | 618 | **0.80** | **0.86** |
| ClinicalBERT | 0.84 | 0.83 | 0.78 | 618 | **0.80** | 0.85 |

Table 11: Performance of models on **WELLXPLAIN** dataset for **SCE**.

| Model | Dimensions | Reservation | Precision | Recall | F-Measure | MCC | Accuracy |
|-------|-----------|-------------|-----------|--------|-----------|-----|----------|
| **BERT** | 6 | 1.00 | 0.72 | 0.62 | 0.64 | 0.55 | 0.87 |
| | | 0.95 | 0.72 | 0.61 | 0.63 | 0.55 | 0.87 |
| | | 0.90 | 0.71 | 0.61 | 0.62 | 0.54 | 0.87 |
| | | 0.85 | 0.71 | 0.60 | 0.62 | 0.54 | 0.87 |
| | | 0.80 | 0.71 | 0.60 | 0.61 | 0.53 | 0.86 |
| | | 0.75 | 0.73 | 0.60 | 0.60 | 0.54 | 0.86 |
| | 5 | 1.00 | 0.80 | 0.74 | 0.75 | 0.65 | 0.87 |
| | | 0.95 | 0.80 | 0.74 | 0.75 | 0.65 | 0.87 |
| | | 0.90 | 0.80 | 0.73 | 0.75 | 0.65 | 0.87 |
| | | 0.85 | 0.80 | 0.73 | 0.75 | 0.65 | 0.87 |
| | | 0.80 | 0.81 | 0.73 | 0.74 | 0.65 | 0.87 |
| | | 0.75 | 0.80 | 0.72 | 0.74 | 0.65 | 0.86 |
| | 4 | 1.00 | 0.83 | 0.81 | 0.82 | 0.72 | 0.90 |
| | | 0.95 | 0.83 | 0.81 | 0.81 | 0.72 | 0.90 |
| | | 0.90 | 0.83 | 0.80 | 0.81 | 0.72 | 0.90 |
| | | 0.85 | 0.83 | 0.81 | 0.81 | 0.72 | 0.90 |
| | | 0.80 | 0.83 | 0.80 | 0.81 | 0.72 | 0.90 |
| | | 0.75 | 0.83 | 0.80 | 0.81 | 0.72 | 0.90 |
| | 3 | 1.00 | 0.64 | 0.64 | 0.59 | 0.26 | 0.58 |
| | | 0.95 | 0.63 | 0.63 | 0.57 | 0.27 | 0.58 |
| | | 0.90 | 0.62 | 0.63 | 0.55 | 0.28 | 0.57 |
| | | 0.85 | 0.61 | 0.62 | 0.52 | 0.28 | 0.56 |
| | | 0.80 | 0.60 | 0.60 | 0.48 | 0.27 | 0.54 |
| | | 0.75 | 0.60 | 0.60 | 0.43 | 0.30 | 0.53 |
| **ClinicalBERT** | 6 | 1.00 | 0.67 | 0.60 | 0.62 | 0.51 | 0.86 |
| | | 0.95 | 0.67 | 0.60 | 0.62 | 0.51 | 0.86 |
| | | 0.90 | 0.66 | 0.60 | 0.62 | 0.51 | 0.86 |
| | | 0.85 | 0.67 | 0.60 | 0.62 | 0.51 | 0.86 |
| | | 0.80 | 0.67 | 0.60 | 0.61 | 0.51 | 0.86 |
| | | 0.75 | 0.68 | 0.60 | 0.61 | 0.52 | 0.86 |
| | 5 | 1.00 | 0.77 | 0.71 | 0.73 | 0.61 | 0.85 |
| | | 0.95 | 0.77 | 0.71 | 0.72 | 0.61 | 0.85 |
| | | 0.90 | 0.77 | 0.71 | 0.72 | 0.60 | 0.77 |
| | | 0.85 | 0.77 | 0.70 | 0.71 | 0.60 | 0.85 |
| | | 0.80 | 0.77 | 0.70 | 0.71 | 0.60 | 0.85 |
| | | 0.75 | 0.76 | 0.69 | 0.70 | 0.60 | 0.84 |
| | 4 | 1.00 | 0.83 | 0.75 | 0.77 | 0.66 | 0.88 |
| | | 0.95 | 0.83 | 0.75 | 0.77 | 0.66 | 0.88 |
| | | 0.90 | 0.83 | 0.75 | 0.77 | 0.66 | 0.88 |
| | | 0.85 | 0.83 | 0.75 | 0.77 | 0.66 | 0.88 |
| | | 0.80 | 0.83 | 0.74 | 0.76 | 0.65 | 0.87 |
| | | 0.75 | 0.82 | 0.74 | 0.76 | 0.65 | 0.87 |
| | 3 | 1.00 | 0.64 | 0.65 | 0.58 | 0.27 | 0.58 |
| | | 0.95 | 0.63 | 0.64 | 0.56 | 0.28 | 0.57 |
| | | 0.90 | 0.63 | 0.63 | 0.54 | 0.29 | 0.56 |
| | | 0.85 | 0.62 | 0.63 | 0.51 | 0.30 | 0.56 |
| | | 0.80 | 0.61 | 0.62 | 0.47 | 0.30 | 0.55 |
| | | 0.75 | 0.61 | 0.62 | 0.43 | 0.32 | 0.53 |

Table 12: Performance of **BERT** and **ClinicalBERT** on MULTIWD dataset for **GL** across various dimensionalities and reservations.

| Model | Dimensions | Reservation | Precision | Recall | F-Measure | MCC | Accuracy |
|---|---|---|---|---|---|---|---|
| **ERNIE** | 6 | 1.00 | 0.76 | 0.68 | 0.71 | 0.62 | 0.88 |
| | | 0.95 | 0.76 | 0.68 | 0.71 | 0.62 | 0.88 |
| | | 0.90 | 0.76 | 0.67 | 0.70 | 0.62 | 0.88 |
| | | 0.85 | 0.76 | 0.67 | 0.70 | 0.61 | 0.88 |
| | | 0.80 | 0.75 | 0.67 | 0.69 | 0.61 | 0.88 |
| | | 0.75 | 0.76 | 0.66 | 0.68 | 0.61 | 0.87 |
| | 5 | 1.00 | 0.83 | 0.77 | 0.79 | 0.69 | 0.88 |
| | | 0.95 | 0.83 | 0.76 | 0.78 | 0.69 | 0.88 |
| | | 0.90 | 0.83 | 0.76 | 0.78 | 0.68 | 0.88 |
| | | 0.85 | 0.83 | 0.76 | 0.78 | 0.69 | 0.88 |
| | | 0.80 | 0.83 | 0.75 | 0.77 | 0.68 | 0.88 |
| | | 0.75 | 0.83 | 0.75 | 0.77 | 0.68 | 0.88 |
| | 4 | 1.00 | 0.87 | 0.83 | 0.84 | 0.75 | 0.91 |
| | | 0.95 | 0.87 | 0.83 | 0.84 | 0.75 | 0.91 |
| | | 0.90 | 0.87 | 0.82 | 0.84 | 0.75 | 0.91 |
| | | 0.85 | 0.87 | 0.82 | 0.84 | 0.75 | 0.91 |
| | | 0.80 | 0.87 | 0.82 | 0.83 | 0.75 | 0.91 |
| | | 0.75 | 0.87 | 0.82 | 0.84 | 0.75 | 0.91 |
| | 3 | 1.00 | 0.63 | 0.64 | 0.59 | 0.26 | 0.58 |
| | | 0.95 | 0.63 | 0.63 | 0.57 | 0.27 | 0.58 |
| | | 0.90 | 0.62 | 0.63 | 0.54 | 0.27 | 0.57 |
| | | 0.85 | 0.61 | 0.62 | 0.52 | 0.28 | 0.56 |
| | | 0.80 | 0.59 | 0.60 | 0.47 | 0.28 | 0.54 |
| | | 0.75 | 0.59 | 0.60 | 0.43 | 0.30 | 0.53 |
| **MentalBERT** | 6 | 1.00 | 0.74 | 0.66 | 0.68 | 0.59 | 0.88 |
| | | 0.95 | 0.74 | 0.66 | 0.67 | 0.59 | 0.88 |
| | | 0.90 | 0.74 | 0.65 | 0.67 | 0.58 | 0.88 |
| | | 0.85 | 0.74 | 0.64 | 0.66 | 0.58 | 0.88 |
| | | 0.80 | 0.73 | 0.64 | 0.66 | 0.58 | 0.87 |
| | | 0.75 | 0.75 | 0.63 | 0.65 | 0.58 | 0.87 |
| | 5 | 1.00 | 0.82 | 0.76 | 0.78 | 0.68 | 0.88 |
| | | 0.95 | 0.82 | 0.76 | 0.77 | 0.68 | 0.88 |
| | | 0.90 | 0.82 | 0.75 | 0.77 | 0.67 | 0.88 |
| | | 0.85 | 0.82 | 0.75 | 0.77 | 0.67 | 0.87 |
| | | 0.80 | 0.82 | 0.74 | 0.77 | 0.67 | 0.87 |
| | | 0.75 | 0.82 | 0.74 | 0.76 | 0.67 | 0.87 |
| | 4 | 1.00 | 0.85 | 0.82 | 0.83 | 0.72 | 0.91 |
| | | 0.95 | 0.84 | 0.81 | 0.82 | 0.72 | 0.90 |
| | | 0.90 | 0.84 | 0.81 | 0.82 | 0.72 | 0.90 |
| | | 0.85 | 0.84 | 0.81 | 0.82 | 0.72 | 0.90 |
| | | 0.80 | 0.84 | 0.80 | 0.82 | 0.72 | 0.90 |
| | | 0.75 | 0.85 | 0.80 | 0.82 | 0.72 | 0.90 |
| | 3 | 1.00 | 0.64 | 0.65 | 0.60 | 0.27 | 0.59 |
| | | 0.95 | 0.63 | 0.64 | 0.58 | 0.27 | 0.58 |
| | | 0.90 | 0.63 | 0.64 | 0.56 | 0.28 | 0.58 |
| | | 0.85 | 0.62 | 0.62 | 0.53 | 0.20 | 0.56 |
| | | 0.80 | 0.60 | 0.61 | 0.48 | 0.28 | 0.55 |
| | | 0.75 | 0.61 | 0.61 | 0.44 | 0.30 | 0.54 |

Table 13: Performance of **ERNIE** and **MentalBERT** on MULTIWD Dataset for **GL** across various dimensionalities and reservations.

| Model | Dimensions | Reservation | Precision | Recall | F-Measure | MCC | Accuracy |
|---|---|---|---|---|---|---|---|
| PsychBERT | 6 | 1.00 | 0.71 | 0.63 | 0.65 | 0.55 | 0.87 |
| | | 0.95 | 0.71 | 0.62 | 0.64 | 0.55 | 0.86 |
| | | 0.90 | 0.71 | 0.62 | 0.64 | 0.54 | 0.86 |
| | | 0.85 | 0.71 | 0.61 | 0.63 | 0.54 | 0.86 |
| | | 0.80 | 0.71 | 0.61 | 0.63 | 0.54 | 0.86 |
| | | 0.75 | 0.71 | 0.61 | 0.62 | 0.54 | 0.86 |
| | 5 | 1.00 | 0.79 | 0.74 | 0.75 | 0.65 | 0.87 |
| | | 0.95 | 0.78 | 0.74 | 0.75 | 0.65 | 0.87 |
| | | 0.90 | 0.78 | 0.73 | 0.75 | 0.64 | 0.87 |
| | | 0.85 | 0.78 | 0.73 | 0.75 | 0.65 | 0.87 |
| | | 0.80 | 0.79 | 0.73 | 0.75 | 0.65 | 0.87 |
| | | 0.75 | 0.79 | 0.73 | 0.75 | 0.65 | 0.87 |
| | 4 | 1.00 | 0.83 | 0.81 | 0.81 | 0.71 | 0.90 |
| | | 0.95 | 0.83 | 0.81 | 0.81 | 0.71 | 0.90 |
| | | 0.90 | 0.83 | 0.80 | 0.81 | 0.71 | 0.90 |
| | | 0.85 | 0.82 | 0.80 | 0.81 | 0.71 | 0.89 |
| | | 0.80 | 0.82 | 0.80 | 0.81 | 0.71 | 0.89 |
| | | 0.75 | 0.82 | 0.80 | 0.81 | 0.71 | 0.89 |
| | 3 | 1.00 | 0.65 | 0.65 | 0.60 | 0.28 | 0.59 |
| | | 0.95 | 0.64 | 0.64 | 0.58 | 0.29 | 0.59 |
| | | 0.90 | 0.63 | 0.63 | 0.55 | 0.29 | 0.57 |
| | | 0.85 | 0.62 | 0.63 | 0.53 | 0.30 | 0.57 |
| | | 0.80 | 0.60 | 0.61 | 0.48 | 0.30 | 0.55 |
| | | 0.75 | 0.61 | 0.61 | 0.44 | 0.32 | 0.54 |
| RoBERTa | 6 | 1.00 | 0.76 | 0.69 | 0.71 | 0.63 | 0.89 |
| | | 0.95 | 0.76 | 0.69 | 0.71 | 0.63 | 0.89 |
| | | 0.90 | 0.75 | 0.68 | 0.70 | 0.62 | 0.88 |
| | | 0.85 | 0.76 | 0.67 | 0.70 | 0.62 | 0.88 |
| | | 0.80 | 0.76 | 0.68 | 0.70 | 0.62 | 0.88 |
| | | 0.75 | 0.76 | 0.67 | 0.69 | 0.61 | 0.88 |
| | 5 | 1.00 | 0.83 | 0.77 | 0.79 | 0.70 | 0.88 |
| | | 0.95 | 0.83 | 0.77 | 0.79 | 0.70 | 0.88 |
| | | 0.90 | 0.83 | 0.76 | 0.78 | 0.69 | 0.88 |
| | | 0.85 | 0.83 | 0.76 | 0.78 | 0.69 | 0.88 |
| | | 0.80 | 0.83 | 0.76 | 0.78 | 0.69 | 0.88 |
| | | 0.75 | 0.82 | 0.75 | 0.77 | 0.68 | 0.87 |
| | 4 | 1.00 | 0.86 | 0.81 | 0.83 | 0.72 | 0.90 |
| | | 0.95 | 0.85 | 0.81 | 0.82 | 0.72 | 0.90 |
| | | 0.90 | 0.85 | 0.80 | 0.82 | 0.72 | 0.90 |
| | | 0.85 | 0.85 | 0.80 | 0.82 | 0.72 | 0.90 |
| | | 0.80 | 0.85 | 0.80 | 0.81 | 0.71 | 0.90 |
| | | 0.75 | 0.85 | 0.79 | 0.81 | 0.72 | 0.90 |
| | 3 | 1.00 | 0.64 | 0.65 | 0.60 | 0.27 | 0.59 |
| | | 0.95 | 0.63 | 0.64 | 0.58 | 0.28 | 0.58 |
| | | 0.90 | 0.63 | 0.63 | 0.56 | 0.28 | 0.58 |
| | | 0.85 | 0.62 | 0.62 | 0.53 | 0.29 | 0.56 |
| | | 0.80 | 0.60 | 0.60 | 0.48 | 0.29 | 0.55 |
| | | 0.75 | 0.60 | 0.60 | 0.44 | 0.30 | 0.54 |

Table 14: Performance of **PsychBERT** and **RoBERTa** on **MULTIWD** dataset for **GL** across various dimensionalities and reservations.

| Model | Precision | Recall | F-Measure | MCC | Accuracy | Reservation |
|---|---|---|---|---|---|---|
| | 0.84 | 0.85 | 0.82 | 0.79 | 0.92 | 100% |
| | 0.83 | 0.85 | 0.81 | 0.78 | 0.92 | 95% |
| BERT | 0.81 | 0.85 | 0.79 | 0.77 | 0.92 | 90% |
| | 0.76 | 0.86 | 0.74 | 0.73 | 0.92 | 85% |
| | 0.69 | 0.69 | 0.64 | 0.63 | 0.92 | 80% |
| | 0.68 | 0.61 | 0.62 | 0.6 | 0.92 | 75% |
| | 0.8 | 0.78 | 0.71 | 0.69 | 0.89 | 100% |
| | 0.79 | 0.78 | 0.70 | 0.68 | 0.89 | 95% |
| ClinicalBERT | 0.76 | 0.78 | 0.70 | 0.67 | 0.88 | 90% |
| | 0.71 | 0.78 | 0.62 | 0.62 | 0.87 | 85% |
| | 0.66 | 0.61 | 0.54 | 0.53 | 0.87 | 80% |
| | 0.65 | 0.54 | 0.52 | 0.51 | 0.87 | 75% |
| | 0.86 | 0.85 | 0.83 | 0.8 | 0.93 | 100% |
| | 0.85 | 0.86 | 0.83 | 0.8 | 0.93 | 95% |
| ERNIE | 0.82 | 0.86 | 0.81 | 0.78 | 0.92 | 90% |
| | 0.77 | 0.86 | 0.76 | 0.74 | 0.92 | 85% |
| | 0.70 | 0.70 | 0.66 | 0.64 | 0.92 | 80% |
| | 0.68 | 0.62 | 0.63 | 0.61 | 0.92 | 75% |
| | 0.86 | 0.85 | 0.82 | 0.79 | 0.93 | 100% |
| | 0.84 | 0.86 | 0.81 | 0.79 | 0.92 | 95% |
| MentalBERT | 0.82 | 0.86 | 0.79 | 0.77 | 0.92 | 90% |
| | 0.76 | 0.86 | 0.73 | 0.73 | 0.91 | 85% |
| | 0.70 | 0.69 | 0.64 | 0.63 | 0.92 | 80% |
| | 0.69 | 0.61 | 0.62 | 0.6 | 0.91 | 75% |
| | 0.82 | 0.81 | 0.78 | 0.75 | 0.91 | 100% |
| | 0.81 | 0.81 | 0.77 | 0.74 | 0.91 | 95% |
| PsychBERT | 0.79 | 0.81 | 0.75 | 0.73 | 0.91 | 90% |
| | 0.74 | 0.82 | 0.70 | 0.69 | 0.9 | 85% |
| | 0.69 | 0.67 | 0.61 | 0.6 | 0.9 | 80% |
| | 0.68 | 0.59 | 0.6 | 0.58 | 0.9 | 75% |
| | 0.86 | 0.85 | 0.84 | 0.81 | 0.93 | 100% |
| | 0.85 | 0.85 | 0.83 | 0.8 | 0.93 | 95% |
| RoBERTa | 0.83 | 0.86 | 0.82 | 0.79 | 0.93 | 90% |
| | 0.78 | 0.87 | 0.77 | 0.76 | 0.93 | 85% |
| | 0.70 | 0.70 | 0.66 | 0.64 | 0.93 | 80% |
| | 0.69 | 0.63 | 0.64 | 0.62 | 0.93 | 75% |
| | 0.85 | 0.85 | 0.82 | 0.8 | 0.93 | 100% |
| | 0.84 | 0.85 | 0.82 | 0.79 | 0.92 | 95% |
| XLNET | 0.81 | 0.85 | 0.8 | 0.77 | 0.92 | 90% |
| | 0.76 | 0.85 | 0.74 | 0.73 | 0.92 | 85% |
| | 0.69 | 0.70 | 0.64 | 0.63 | 0.92 | 80% |
| | 0.68 | 0.62 | 0.63 | 0.6 | 0.92 | 75% |

Table 15: Performance of models on **WELLXPLAIN** dataset for **GL** across various reservations.

| Model | Dimensions | Reservation | Precision | Recall | F-Measure | MCC | Accuracy |
|-------|-----------|-------------|-----------|--------|-----------|-----|----------|
| **XLNET** | 6 | 1.00 | 0.74 | 0.66 | 0.68 | 0.59 | 0.88 |
| | | 0.95 | 0.74 | 0.65 | 0.68 | 0.59 | 0.88 |
| | | 0.90 | 0.74 | 0.65 | 0.67 | 0.58 | 0.87 |
| | | 0.85 | 0.74 | 0.64 | 0.67 | 0.57 | 0.87 |
| | | 0.80 | 0.73 | 0.64 | 0.66 | 0.58 | 0.87 |
| | | 0.75 | 0.74 | 0.64 | 0.66 | 0.57 | 0.87 |
| | 5 | 1.00 | 0.81 | 0.75 | 0.77 | 0.67 | 0.88 |
| | | 0.95 | 0.80 | 0.74 | 0.76 | 0.66 | 0.87 |
| | | 0.90 | 0.79 | 0.73 | 0.75 | 0.65 | 0.87 |
| | | 0.85 | 0.79 | 0.73 | 0.75 | 0.65 | 0.87 |
| | | 0.80 | 0.80 | 0.73 | 0.75 | 0.65 | 0.87 |
| | | 0.75 | 0.80 | 0.74 | 0.75 | 0.66 | 0.87 |
| | 4 | 1.00 | 0.85 | 0.82 | 0.83 | 0.73 | 0.91 |
| | | 0.95 | 0.85 | 0.82 | 0.83 | 0.73 | 0.91 |
| | | 0.90 | 0.85 | 0.82 | 0.83 | 0.73 | 0.91 |
| | | 0.85 | 0.85 | 0.81 | 0.82 | 0.73 | 0.91 |
| | | 0.80 | 0.85 | 0.81 | 0.83 | 0.73 | 0.90 |
| | | 0.75 | 0.85 | 0.81 | 0.83 | 0.73 | 0.90 |
| | 3 | 1.00 | 0.64 | 0.65 | 0.59 | 0.27 | 0.59 |
| | | 0.95 | 0.64 | 0.64 | 0.58 | 0.28 | 0.58 |
| | | 0.90 | 0.63 | 0.63 | 0.55 | 0.28 | 0.57 |
| | | 0.85 | 0.62 | 0.63 | 0.52 | 0.29 | 0.56 |
| | | 0.80 | 0.60 | 0.61 | 0.48 | 0.29 | 0.55 |
| | | 0.75 | 0.60 | 0.61 | 0.44 | 0.31 | 0.54 |

Table 16: Performance of **XLNET** on **MULTIWD** dataset for **GL** across various dimensionalities and reservations.

**Prompt:** "First, understand the following definitions: Physical Aspect (PA): Physical wellness fosters healthy dietary practices while discouraging harmful behaviors like tobacco use, drug misuse, and excessive alcohol consumption. Achieving optimal physical wellness involves regular physical activity, sufficient sleep, vitality, enthusiasm, and beneficial eating habits. Body shaming can negatively affect physical well-being by increasing awareness of medical history and appearance issues. Intellectual Aspect (IA): Utilizing intellectual and cultural activities, both inside and outside the classroom, and leveraging human and learning resources enhance the wellness of an individual by nurturing intellectual growth and stimulation. Vocational Aspect (VA): The Vocational Dimension acknowledges the role of personal gratification and enrichment derived from one's occupation in shaping life satisfaction. It influences an individual's perspective on creative problem-solving, professional development, and the management of financial obligations. Social Aspect (SA): The Social Dimension highlights the interplay between society and the natural environment, increasing individuals' awareness of their role in society and their impact on ecosystems. Social bonds enhance interpersonal traits, enabling a better understanding and appreciation of cultural influences. Spiritual Aspect (SpA): The Spiritual Dimension involves seeking the meaning and purpose of human life, appreciating its vastness and natural forces, and achieving harmony within oneself. Emotional Aspect (EA): The Emotional Dimension enhances self-awareness and positivity, promoting better emotional control, realistic self-appraisal, independence, and effective stress management.

Now, you will be given a textual post. Classify the post into one of these labels: 1, 2, 3, or 4. If the post is physical aspect, return 1; if it is either intellectual or vocational aspect, or both of these aspects, return 2; if the post is social aspect, return 3; and if the post is either spiritual or emotional, or both of these aspect, return 4. Then JUST list the key parts of the post that primarily influenced your prediction. Provide your output as a Python list with two values: the first representing your prediction (1, 2, 3, or 4) and the second representing the most important parts for your prediction like the following.

$$value1, value2$$

Textual post: {post}"

---

**Response:**

Figure 8: Prompt used for zero-shot setting.