# OpenReview forum: "WellDunn: On the Robustness and Explainability of Language Models and Large Language Models in Identifying Wellness Dimensions"
_EMNLP/2024/Workshop/BlackBoxNLP — BlackboxNLP 2024_

### Official Review · Reviewer_HAfF · 2024-09-07

**Overall Assessment:** 3
**Confidence:** 3

**Best Paper:**

1

**Best Paper Justification:**

na

**Comments Questions Suggestions And Typos:**

NA

**Paper Summary:**

This study explores the use of Language Models (LMs) in mental health applications, where predictive performance alone may not suffice for clinical practice. A key issue is whether these models' explanations align with clinical determinations. The research evaluates the robustness and explainability of LMs in identifying Wellness Dimensions (WD) using two mental health datasets.

**Summary Of Strengths:**

Two datasets have been used to  study the use LMs in mental health
The study focuses on the model’s attention and confidence in predicting WDs.
LLM performance on zero-shot and few-shot,  prompting, and fine-tuning is explored.
Domain specific models are trained as well as general ones.

**Summary Of Weaknesses:**

Only 1 medical LLM is trained that performs worse than LLAMA which is expected given LLAMA is trained on more data.
Robustness is only measured through loss and explainability through a score.

---

### Official Review · Reviewer_3N9N · 2024-09-07

**Overall Assessment:** 3
**Confidence:** 2

**Best Paper:**

1

**Best Paper Justification:**

None

**Comments Questions Suggestions And Typos:**

Table 1's subheadings are not aligned.

**Paper Summary:**

This paper focuses on the alignment of the Language Models explanations and clinical determination. They propose an evaluation framework called WellDunn that emphasizes the robustness and explainability of language models (LMs) in identifying Wellness Dimensions (WD). They propose evaluation schemes examining the consistency in LM’s attention with ground-truth explanations and confidence in predictions.

Dataset: Multi-label Classification-based MultiWD, WellXplain

Evaluation Welldunn Evaluation Criteria:
Robustness: Sigmoid Cross Entropy(SCE) and Gambler’s Loss(GL).
Explainability: Singular Value Decomposition (SVD) and Attention-Overlap (AO) Score.

Findings:
1) Across all LMs/LLMs, the alignment between attention and explanations remains low
2) GPT-3.5/4 lag behind RoBERTa, and MedAlpaca, a fine-tuned LLM on wellxplain fails to deliver any remarkable improvements in performance or explanations
3) Re-examining LMs' predictions based on a confidence-oriented loss function reveals a significant performance drop.

**Summary Of Strengths:**

Things that I like in this paper:
1. This paper is very well motivated and really well written.
2. Their study is comprehensive. The study examines multiple language models (both general-purpose and domain-specific) and large language models, providing a thorough comparison of their performance.
3. This paper provide some insightful error analysis.

**Summary Of Weaknesses:**

1. The paper notes that Gambler's Loss, intended to make models more risk-averse, led to unexpected performance drops in some cases. This inconsistency is not fully explained.
2. The AO score is always so low. I am not sure that whether it is just because using Attention as a medium of explanation is not a good idea. The paper does not talk about how reliable is the Attention-Overlap (AO) score as a measure of explanation quality. Also I think maybe the low AO scores be due to limitations in the ground truth explanations rather than model shortcomings.
3. The paper does not explain how does the choice of top-k tokens (currently top 4) in calculating the AO score affect the results.
4. The paper does not compare to other popular explanation methods.
5. The paper does not address whether attention patterns are consistent across different runs or initializations, which could affect the reliability of the AO score.

---

### Official Review · Reviewer_hZ1v · 2024-09-12

**Overall Assessment:** 3
**Confidence:** 2

**Best Paper:**

1

**Comments Questions Suggestions And Typos:**

The first research question line 365 "Does the performance of LMs depend on the number of WDs in the datasets" is a bit strange since the accuracy will always increase when classes are combined. So the answer to the question is known in advance. Perhaps the authors want to rephrase the question to - how much this will affect accuracy.

Figure 2 is not clear. What do the rectangles mean? what does the "in"/"out"

Although there is a reference to related works, the review lacks reference to works that are similar and close. For example, the authors refer to works related to mental health conditions, however, their work is related to many other aspects e.g., emotions and well-being and explainability. And there is a wide literature on each of these topics

**Paper Summary:**

This paper, "WellDunn: On the Robustness and Explainability of Language Models and Large Language Models in Identifying Wellness Dimensions", suggests a new method for solving existing dataset (MULTIWD, and WELLXPLAIN). The dataset includes reddit posts annotated by experts (clinical domain for mental health) and includes both classification class and a highlight of the text that "explains" the classification decision.

The method that the author suggests involves a type of fine-tuning with a specific loss function (Gambler’s loss function, standard sigmoid cross entropy loss)  and by using attention mechanics.

The authors measure the performance of their method compared to other of shelf LLMs and domain-specific LMs.

The authors find their method to beat all the other methods.

**Summary Of Strengths:**

There is an overview of a variety of models dedicated to the task.

The method that the author proposed seems to be novel for the relevant task and beats the other methods. I was impressed by their surprising finding which shows that their new technique can bring much better performance (accuracy) than using SOTA models (i.e., GPT4 and more). This is also a gap with previous literature. The best accuracy score reported by previous literature was 41.55 (Sathvik and Muskan Garg. 2023) while the authors achieved F1 0.82.

The subject is important and interesting and has an applicative value and the researchers have performed many experiments that the knowledge can be used by others.

**Summary Of Weaknesses:**

It was hard for me to follow the details of the experiment and models' architecture, and unfortunately, I couldn't understand exactly what the method was. In particular, I did not understand the paragraph "Making LMs risk-averse with abstention" and also did not follow "Evaluation Strategy".

Re the statement "LLMs’ internal machinery is not as transparent as LMs" there are already open source LLMs such as Llama70b
The authors used orca_mini_3b which is a smaller version and there are much more.

The decrease in the performance of GPT4 and Llama as more examples are added is unlikely. How do the authors explain it?

---

### Decision · Program_Chairs · 2024-09-19

**Decision:**

Accept

**Comment:**

While the reviews were largely ambivalent, I think that the ideas and findings in this paper outweigh the weaknesses pointed out by reviewers. The findings point out the importance of evaluating models not just in terms of their behaviors, but also in terms of the factors that led to particular decisions; this is a good fit for the themes of the workshop. While the methods used to uncover these important features may not themselves be robust (as some reviewers' concerns hint at), I think this will be a good topic of discussion at the workshop. One thing that will be especially important to address for the camera-ready will be better contextualizing this work with relevant related work, as pointed out by Reviewers hZ1v and 3N9N. It will also be helpful to comparing against more relevant baselines that are designed for this particular task setting.